# Synthesis and Self-Assembly of Hyperbranched Multiarm Copolymer Lysozyme Conjugates Based on Light-Induced Metal-Free Atrp

**DOI:** 10.3390/nano13061017

**Published:** 2023-03-11

**Authors:** Jianguo Yi, Yan Qin, Yue Zhang

**Affiliations:** 1School of Chemical Engineering and Technology, Hebei University of Technology, Tianjin 300130, China; 2Hebei Key Laboratory of Functional Polymers, Tianjin 300130, China

**Keywords:** hyperbranched polymers, lysozyme, copolymer protein conjugates, temperature responsiveness, self-assembly

## Abstract

In recent years, the coupling of structurally and functionally controllable polymers with biologically active protein materials to obtain polymer–protein conjugates with excellent overall properties and good biocompatibility has been important research in the field of polymers. In this study, the hyperbranched polymer hP(DEGMA-*co*-OEGMA) was first prepared by combining self-condensation vinyl polymerization (SCVP) with photo-induced metal-free atom transfer radical polymerization (ATRP), with 2-(2-bromo-2-methylpropanoyloxy) ethyl methacrylate (BMA) as inimer, and Di (ethylene glycol) methyl ether methacrylate (DEGMA) and (oligoethylene glycol) methacrylate (OEGMA, *M*_n_ = 300) as the copolymer monomer. Then, hP(DEGMA-*co*-OEGMA) was used as a macroinitiator to continue the polymerization of a segment of pyridyl disulfide ethyl methacrylate (DSMA) monomer to obtain the hyperbranched multiarm copolymers hP(DEGMA-*co*-OEGMA)-*star*-PDSMA. Finally, the lysozyme with sulfhydryl groups was affixed to the hyperbranched multiarm copolymers by the exchange reaction between sulfhydryl groups and disulfide bonds to obtain the copolymer protein conjugates hP(DEGMA-*co*-OEGMA)-*star*-PLZ. Three hyperbranched multiarm copolymers with relatively close molecular weights but different degrees of branching were prepared, and all three conjugates could self-assemble to form nanoscale vesicle assemblies with narrow dispersion. The biological activity and secondary structure of lysozyme on the assemblies remained essentially unchanged.

## 1. Introduction

Lysozyme is an alkaline, naturally occurring antibacterial protein that is widely found in animal egg whites and in a variety of human tissues [1,2]. Lysozyme achieves the breakdown of bacteria by breaking the β-1,4 glycosidic bond in the cell wall that connects N-acetyl amino acids to N-acetyl glucosamine [3]. The immobilization of such functional proteins such as lysozyme conjugated to polymers can be applied in different industries such as medicine and the food industry [4,5,6,7]. In previous studies, proteins were usually modified onto linear polymers to obtain bioconjugates. For example, Tao and colleagues prepared linear and branched poly (N-(2-hydroxypropyl) methacrylamide) lysozyme bioconjugate, respectively, and studied and compared their biological activities [8]. Compared with linear polymers, hyperbranched polymers have attracted great interest due to their inherent structure with many excellent properties, such as higher solubility, lower viscosity and a higher level of terminal functional groups. There are three main methods for the synthesis of hyperbranched polymers, the most commonly used being the self-condensation vinyl polymerization (SCVP) method [9,10,11,12,13,14,15,16,17,18,19,20]. To control the reaction process and polymer molecular weight, the SCVP method is usually combined with controlled radical polymerization. For example, Patrickios and colleagues prepared hydrophobic, degradable hyperbranched polymers PMMA by combining the SCVP method with atom transfer radical polymerization (ATRP), and obtained amphiphilic hyperbranched dobby polymers by using PMMA as a macromolecular initiator, followed by polymerization of 2-(dimethylamino) ethyl methacrylate (DMAEMA) [21]. However, for various applications such as microelectronics and biomaterials, a key limitation of using ATRP to prepare polymers is metal contamination, and it is preferable to use metal-free catalysis to prepare hyperbranched polymers. Hawker’s group reported a metal-free ATRP process catalyzed by the light-induced organic molecule 10-phenylphenothiazine (PTH), where vinyl monomers can be efficiently activated and de-activated, allowing good control of the molecular weight of the resulting polymers, e.g., MMA, BnMA, DMAEMA, etc. [22]. Yagci and his colleagues prepared lightly branched, hyperbranched and cross-linked methyl methacrylate and styrene polymers by light-induced metal-free ATRP with perylene as the photocatalyst [23]. Here, there is much interest in using the excellent properties of hyperbranched polymers in combination with functional proteins. In the preparation process, the exchange reaction of sulfhydryl [24] groups with disulfide bonds is crucial. Bontempo and his colleagues prepared a poly (hydroxyethyl methacrylate) containing a disulfide dipyridyl group at the end by ATRP polymerization, and received the protein (BSA) onto the polymer by the exchange reaction of sulfhydryl groups and disulfide esters [25]. This method has mild reaction conditions and is suitable for the preparation of various bioconjugates.

In this study, hyperbranched multiarm copolymers hP(DEGMA-*co*-OEGMA)-*star*-PDSMA were prepared by a light-induced metal-free ATRP method. Firstly, the hyperbranched polymers were obtained by polymerizing Di (ethylene glycol) methyl ether methacrylate (DEGMA) and (oligoethylene glycol) methacrylate (OEGMA, *M*_n_ = 300) with different monomer ratios under UV lamp irradiation at 380 nm using PTH as the photocatalyst and 2-(2-bromo-2-methylpropanoyloxy) ethyl methacrylate (BMA) as the inimer. The hyperbranched multiarm copolymers hP(DEGMA-*co*-OEGMA)-*star*-PDSMA was then obtained by continuing the polymerization of a segment of pyridyl disulfide ethyl methacrylate (DSMA) monomer at its end using hP(DEGMA-*co*-OEGMA) as a macroinitiator. The lysozyme was modified with a sulfhydryl group on the lysine residue of the lysozyme by diiminothiane hydrochloride (Traut’s) reagent, and the lysozyme was conjugated to hP(DEGMA-*co*-OEGMA)-*star*-PDSMA with a hyperbranched core using a sulfhydryl disulfide bond exchange reaction to obtain the protein-modified bioconjugate hP(DEGMA-*co*-OEGMA)-*star*-PLZ (Figure 1). We prepared three hyperbranched multiarm copolymers with close molecular weights and different branching degrees, and investigated the self-assembly behavior of the three bioconjugates, i.e., the morphology and size of the assemblies above LCST, and the biological activity of lysozyme on the bioconjugate. With the invocation of functional proteins in hyperbranched dobby copolymers, the self-assemblies [26,27,28,29,30,31] are expected to be applied as nanocarriers.

## 2. Experimental Section Materials

Di (ethylene glycol) methyl ether methacrylate (DEGMA, *M*_n_ = 188, Sigma-Aldrich, St. Louis, Missouri, MI, USA, 99%), (oligoethylene glycol) methacrylate (OEGMA, *M*_n_ = 300, Sigma-Aldrich, St. Louis, Missouri, MI, USA, 99%) and 2-hydroxyethyl methacrylate (Sigma-Aldrich, St. Louis, Missouri, MI, USA, 99%) were removed by alkaline alumina silica columns polymerization inhibitor. 2-Bromoisobutyryl bromide (Tianjin Guangfu Chemical Reagent Technology Co., Ltd., Tianjin, China, 98%). 2,2′-dithiodipyridine (HEOWNS, Tianjin, China, 99%). 2-mercaptoethanol (Tianjin Guangfu Fine Chemical, Tianjin, China, 98%). Methacryloyl chloride (HEOWNS, Tianjin, China, 98%). Lysozyme (Sigma-Aldrich, St. Louis, Missouri, MI, USA) Diiminothiane hydrochloride (Traut’s reagent, Sigma-Aldrich, St. Louis, Missouri, MI, USA, 98%). DL-Dithiothreitol (Sigma-Aldrich, St. Louis, Missouri, MI, USA, 99%). Ellman’s reagent (DTNB, 99%, Alfa Aesar, Shanghai, China). Glycerol. Micrococcus lysis cells (Tianjin Dingguo Biotechnology, Tianjin, China). All solvents were distilled after desiccant drying prior to use.

### 2.1. Synthesis of Photocatalyst 10-Phenylphenothiazine (PTH)

NaOtBu (134 mg, 1.40 mmol), phenothiazine (199 mg, 1.00 mmol), RuPhos Precat (14 mg, 0.02 mmol, 2 mol%) and RuPhos (8 mg, 0.02 mmol, 2 mol%) were added to the Schlenk flask. The flask was degassed by three freeze-pump-thaw cycles. Then, anhydrous dioxane (1 mL) was added under nitrogen protection. Finally, anhydrous chlorobenzene (143 μL, 1.40 mmol) was added. The solution was stirred at 110 °C for 5 h, cooled to room temperature after the reaction was completed, diluted with a small amount of dichloromethane, washed three times with distilled water and NaCl solution, and dried with Mg_2_SO_4_. The crude product was purified by silica gel column chromatography, petroleum ether:ethyl acetate = 20:1. (Yield: 97%).

### 2.2. Synthesis of the Inimer BMA

The inimer BMA was prepared according to a previous report. 2-Hydroxyethyl methacrylate (HEMA) (6.0 mL, 50 mmol) and triethylamine (15.0 mL, 108 mmol) were dissolved in 50 mL of dry dichloromethane, then cooled to 0 °C with an ice-water bath. An amount of 10 mL of a dichloromethane solution containing 2-bromoisobutylacyl bromide (7.0 mL, 57 mmol) was slowly added dropwise to the solution. After the dropwise addition was complete, the solution was stirred at room temperature for 24 h. After the reaction, the salt produced by the reaction was removed by suction filtration, the filtrate was washed three times with NaHCO_3_ solution, dried with Na_2_SO_4_ and the remaining filtrate was concentrated. The crude product was purified by silica gel column chromatography, petroleum ether:ethyl acetate = 5:1. (Yield: 80%).

### 2.3. Synthesis of DSMA

Hydroxyethyl disulfide pyridine (1.50 g, 8.27 mmol) and triethylamine (1.26 g, 12.40 mmol) were dissolved in 30 mL of dry dichloromethane and cooled to 0 °C with an ice-water bath, then 20 mL dichloromethane solution containing methacryloyl chloride (1.30 g, 12.4 mmol) was slowly added dropwise to the solution. After the dropwise addition was completed, the solution was stirred at room temperature for 12 h. After the reaction, the salt produced by the reaction was removed by suction filtration, washed three times with distilled water and saturated NaCl solution, and dried with Na_2_SO_4_. The crude product was purified by silica gel column chromatography, petroleum ether:ethyl acetate = 4:1. (Yield: 75%).

### 2.4. Synthesis of hP(DEGMA-co-OEGMA)

DEGMA, OEGMA (*M*_n_ = 300), BMA and PTH were dissolved in DMF, deaerated and protected by argon atmosphere. The polymerization was irradiated under a UV lamp with a wavelength of 380 nm and stirred at room temperature for 12 h. The polymer was purified by precipitation twice in diethyl ether. The details are shown in Table 1.

### 2.5. Synthesis of hP(DEGMA-co-OEGMA)-star-PDSMA

hP(DEGMA-*co*-OEGMA), DSMA and PTH were dissolved in DMF, deaerated and protected by argon atmosphere. The polymerization was irradiated under a UV lamp with a wavelength of 380 nm and stirred at room temperature for 1 h. The polymer was purified by precipitation twice in diethyl ether. The details are shown in Table 2.

### 2.6. Synthesis of Thiol-Modified Lysozyme

Lysozyme (100 mg, 0.00680 mmol) was dissolved in 24 mL of phosphate buffered solution (PBS, pH = 8.0, 10 mM), the solution was degassed by three freeze-pump-thaw cycles. Under the protection of argon atmosphere, Traut’s reagent solution (3.1 mg, 0.022 mmol, dissolved in 1 mL of phosphate buffer solution) was added dropwise to the lysozyme solution, stirred under an ice bath for 72 h, and the degree of reaction was detected by Ellman’s reagent (DTNB).

### 2.7. Preparation of Bioconjugate

The lysozyme modified with sulfhydryl groups (4.0 mg) was dissolved in 4.0 mL PBS and the solution was protected by argon atmosphere. The hyperbranched multiarm copolymers h_1_P(DEGMA-*co*-OEGMA)-*star*-PDSMA (2.0 mg) was dissolved in 0.1 mL DMF and added to the solution under argon protection. The reaction was conducted at room temperature for 12 h. For h_2_P(DEGMA-*co*-OEGMA)-*star*-PDSMA (4.0 mg), h_3_P(DEGMA-*co*-OEGMA)-*star*- PDSMA (4.0 mg) and 4 mL of lysozyme solution at a concentration of 1 mg/mL were added, respectively.

### 2.8. LCST Measurements

The LCST of the hyperbranched polymer and bioconjugate solutions were determined on a UV-Vis spectrophotometer, which is the trend of transmittance at a wavelength of 600 nm with temperature. The concentration of the hyperbranched polymer and bioconjugate solution was 1 mg/mL, and the equilibration time for each temperature change was 3 min, with a temperature change interval of 1 °C.

### 2.9. Self-Assembly of hP(DEGMA-co-OEGMA)-star-PLZ in Water

For hP(DEGMA-*co*-OEGMA)-*star*-PLZ, 0.8 mg/mL aqueous solution was prepared at 8 °C and heated to 45 °C to obtain the self-assemblies. Particle size and particle size distribution of the assemblies were measured by dynamic light scattering (DLS) at 45 °C.

### 2.10. Polyacrylamide Gel Electrophoresis (SDS-PAGE)

SDS-PAGE was performed using a 14% cross-linked polyacrylamide gel and glycine running buffer (20 mM Tris, 1.25 M glycine and 0.1% SDS). Electrophoresis was performed at 68 mV and 16 mA for 2.5 h. The polyacrylamide gel was stained with Coomassie brilliant blue solution and decolorized with a mixture of methanol and glacial acetic acid.

### 2.11. Determination of Lysozyme Bioactivity

Using Micrococcus lysis (ML) cells as the reaction substrate for the activity test of lysozyme and its bioconjugates, 5 mg of ML cells were dissolved in 31 mL of phosphate buffer solution (pH = 6.6, 66 mM), standed overnight in a refrigerator at 4 °C. Before testing the activity, the ML cells solution needs to be shaken on a shaker (25 °C) for 3.5 h. Lysozyme and bioconjugates were dissolved in phosphate buffer solution (pH = 8.0, 10 mM), the concentration of lysozyme is 1 mg/mL. An amount of 175 μL of lysozyme solution/bioconjugates solution was added to 700 μL of ML cells solution at 25 °C, and the change in absorbance at 450 nm was detected by UV-Vis spectrophotometer at intervals of 5 s. The absorbance ratio was used to evaluate the activity of lysozyme.

## 3. Results and Discussion

In a previous study, Hawker and colleagues reported that PTH as a photocatalyst can effectively control the polymerization of DEGMA [22]. In the present study, three temperature-sensitive hyperbranched polymers hP(DEGMA-*co*-OEGMA) with different branching degrees and molecular weights, were prepared using photo-induced metal-free ATRP by adjusting the feed ratios of monomer OEGMA (*M*_n_ = 300), DEGMA and inimer BMA. The ^1^H NMR spectra of PTH and BMA prepared according to the available reports are shown in Appendix A. The feed ratios of DEGMA, OEGMA (*M*_n_ = 300) and BMA were 7:3:1, 5.6:2.4:1 and 3.5:1.5:1, corresponding to h_1_P(DEGMA-*co*-OEGMA), h_2_P(DEGMA-*co*-OEGMA) and h_3_P(DEGMA-*co*-OEGMA). The structures of the hyperbranched polymers were characterized by ^1^H NMR and GPC.

The ^1^H NMR spectrum of the polymer h_1_P(DEGMA-*co*-OEGMA) is shown in Figure 1a. It can be observed that the sharp peaks appearing at 3.42, 3.57–3.71 and 4.14 ppm correspond to the DEGMA and OEGMA (*M*_n_ = 300) main chain C*H_3_*, C*H_2_*O and COOC*H_2_* protons, respectively; in addition, the peak belonging to the C*H_2_* proton in the BMA structure can also be observed at 4.38 ppm. The ^1^H NMR spectra of h_2_P(DEGMA-*co*-OEGMA) and h_3_P(DEGMA-*co*-OEGMA) are shown in Appendix A. Based on the ^1^H NMR results, the DP_n_ ratios of DEGMA, OEGMA (*M*_n_ = 300) and BMA for h_1_P(DEGMA-*co*-OEGMA), h_2_P(DEGMA-*co*-OEGMA) and h_3_P(DEGMA-*co*-OEGMA) can be calculated as 6.8:2.8:1, 5.7:2.2:1 and 3.3:1.4:1, respectively. Since BMA and the copolymer monomers DEGMA and OEGMA (*M*_n_ = 300) are methacrylates, the DP_n_ ratios are close to the feed ratios. The GPC curve of h_1_P(DEGMA-*co*-OEGMA), h_2_P(DEGMA-*co*-OEGMA) and h_3_P(DEGMA-*co*-OEGMA) is shown in Figure 1b, Appendix A. As can be seen in Figure 1b, the GPC curve of h_1_P(DEGMA-*co*-OEGMA) shows two peaks. Fitting the curve (Appendix A), it can be obtained that the Mn of these two peaks is 12.2 and 24.1 kDa, respectively, with a peak area ratio of 91.1:8.9. The higher *M*_n_ is about twice of the lower *M*_n_, which may be due to the coupling between the lower *M*_n_ polymer chains. The ratio of higher *M*_n_ polymers is low due to volume hindrance. A similar phenomenon can be observed in the GPC curves of h_2_P(DEGMA-*co*-OEGMA) and h_3_P(DEGMA-*co*-OEGMA) (Appendix A). It can be found that the percentage of high *M*_n_ polymers increases with the increase in DP_n_ of BMA, which is due to the high percentage of BMA that facilitates the conjugation of polymer chains and thus the coupling. The structural data of the three hyperbranched polymers based on ^1^H NMR and GPC results are summarized in Table 3.

Hyperbranched multiarm copolymers hP(DEGMA-*co*-OEGMA)-*star*-PDSMA with an internal hyperbranched core and external linear PDSMA arms were prepared by continuing the polymerization of DSMA monomer at the end of the hyperbranched polymer hP(DEGMA-*co*-OEGMA). The ^1^H NMR spectrum of DMSA prepared according to the available reports is shown in Appendix A. In this study, these hyperbranched multiarm copolymers were named h_1_P(DEGMA-*co*-OEGMA)-*star*-PDSMA_45_, h_2_P(DEGMA-*co*-OEGMA)-*star*-PDSMA_48_ and h_3_P(DEGMA-*co*-OEGMA)-*star*-PDSMA_54_. Three hyperbranched multiarm copolymers were prepared with h_1_P(DEGMA-*co*-OEGMA), h_2_P(DEGMA-*co*-OEGMA) and h_3_P(DEGMA-*co*-OEGMA) as cores, respectively, and characterized by ^1^H NMR and GPC. The ^1^H NMR spectra of h_1_P(DEGMA-*co*-OEGMA)-*star*-PDSMA_45_ are shown in Figure 1c, where the peaks at 7.12, 7.69 and 8.48 ppm belonging to the pyridine ring on PDSMA can be clearly seen. The ^1^H NMR spectra of h_2_P(DEGMA-*co*-OEGMA)-*star*-PDSMA_48_ and h_3_P(DEGMA-*co*-OEGMA)-*star*-PDSMA_54_ are shown in Appendix A. The DP_n_ of DSMA was calculated separately from the ^1^H NMR spectra. H_1_P(DEGMA-*co*-OEGMA)-*star*-PDSMA_45_ GPC curve is shown as the red line in Figure 1b, which shifts toward higher molecular weight compared to the black line, proving that DSMA was successfully conjugated to the hyperbranched polymer hP(DEGMA-*co*-OEGMA). Additionally, the GPC curve showed two peaks, which were 5.1 and 35.1 kDa for *M*_n_, respectively, according to the fitting results (Appendix A). The percentage of lower molecular weight polymers is quite low (1.2%), indicating that most of the DSMA monomers are polymerized into higher *M*_n_ hyperbranched polymers with more bromine groups. A similar phenomenon can be found in the GPC fitting results for h_2_P(DEGMA-*co*-OEGMA)-*star*-PDSMA_48_ and h_3_P(DEGMA-*co*-OEGMA)-*star*-PDSMA_54_ (Appendix A). The structural data of the three hyperbranched multiarm copolymers based on the ^1^H NMR and GPC results are summarized in Table 4.

The lysozyme modified with sulfhydryl groups was obtained by the reaction of Traut’s reagent with lysozyme. During the reaction, the molar ratio of Traut’s reagent to lysozyme was 3.2:1. The successful modification of lysozyme was demonstrated by UV-Vis spectrophotometry, as shown in Figure 2, the pure Ellman’s reagent had no absorption at 400 nm, while the sulfhydryl modified lysozyme had strong absorption at 400 nm, Ellman’s reagent was used to test the content of sulfhydryl groups on lysozyme, and the average of 0.8 sulfhydryl groups reacted on each lysozyme molecule was obtained by calculation.

The lysozyme modified with sulfhydryl groups was conjugated to the hyperbranched multiarm copolymers hP(DEGMA-*co*-OEGMA)-*star*-PDSMA by an exchange reaction of disulfide bonds between thiol and dithiopyridine groups [32,33]. When the reaction was finished, the solution was heated to 45 °C and the solution appeared turbid, and the supernatant was centrifuged to detect the reaction conversion by using the Bicinchoninic Acid Assay (BCA) method. The standard curve of lysozyme was prepared by measuring different concentration gradients of lysozyme solution using an enzyme standardizer (Appendix A). The standard curve allowed us to calculate the concentration of lysozyme in the supernatant after centrifugation, which in turn gave the concentration of unreacted lysozyme and, therefore, the number of lysozymes grafted on the hyperbranched multiarm copolymers chain. In this study, three bioconjugates were prepared, named h_1_P(DEGMA-*co*-OEGMA)-*star*-PLZ(G-1), h_2_P(DEGMA-*co*-OEGMA)-*star*-PLZ(G-2) and h_3_P(DEGMA-*co*-OEGMA)-*star*-PLZ(G-3), with 8, 10 and 11 graft numbers of lysozyme, respectively.

The synthesis of bioconjugates can also be demonstrated by 14% polyacrylamide gel electrophoresis (SDS-PAGE). Protein bioconjugates prepared by the exchange reaction of sulfhydryl disulfide bonds can use DTT to break the disulfide bonds between proteins and hyperbranched polymers and thus release the proteins for dissociation. As shown in Figure 3, lysozyme (lane 2), sulfhydryl-modified lysozyme (lane 3) and DTT-treated lysozyme (lane 6) produced almost identical bands at 14.4 kDa. Since there was no lysozyme modified at the end of the hyperbranched multiarm copolymers hP(DEGMA-*co*-OEGMA)-*star*-PDSMA, the polymer did not flow on the gel, so no band was observed for lane 5. The protein-hybridized bioconjugates G-1 was unable to flow through the polyacrylamide gel due to its large size, so no band was observed in lane 4. After treatment with DTT, a clear band at 14.4 kDa could be observed (lane 7), indicating that the lysozyme molecule was conjugated to the hyperbranched multiarm copolymers polymer through a disulfide bond. The same results can be found in the SDS-PAGE of G-2 and G-3 (Appendix A).

Polyethylene glycol (PEG) chains exhibit LCST in aqueous solutions due to the balance between the molecules in the hydrophilic and hydrophobic portions of the molecular structure of the hyperbranched polymer hP(DEGMA-*co*-OEGMA) [34,35]. The LCST of the polymer is regulated by adjusting the length of the PEG chains and the ratio of the composition of the hydrophilic and hydrophobic portions of the polymer. The light transmission of the hyperbranched polymer solution at different temperatures is measured by a UV-Vis spectrophotometer. The thermal responsiveness behavior of the polymer was investigated by measuring the transmittance of hyperbranched polymer solutions at different temperatures using a UV-Vis spectrophotometer. The LCST of hyperbranched polymers is defined as the inflection point of the spectral curve. As shown in Figure 4, the light transmittance of aqueous solutions of h_1_P(DEGMA-*co*-OEGMA), h_2_P(DEGMA-*co*-OEGMA) and h_3_P(DEGMA-*co*-OEGMA) hyperbranched polymers decreased significantly at temperatures above 33 °C, 24 °C and 15 °C, respectively, so the LCST of hyperbranched polymers h_1_P(DEGMA-*co*-OEGMA), h_2_P(DEGMA-*co*-OEGMA) and h_3_P(DEGMA-*co*-OEGMA) were 33 °C, 24 °C and 15 °C, respectively. The molar ratios of the three prepared hyperbranched polymers OEGMA (*M*_n_ = 300) and DEGMA are the same, so the larger the proportion of hydrophobic monomer BMA feeding, the more LCST decreases [36]. In addition, the branched structure of the hyperbranched polymers also influenced their thermal responsiveness behavior.

The thermal response behavior of hP(DEGMA-*co*-OEGMA)-*star*-PDSMA was not measured because the PDSMA homopolymer is hydrophobic. The transmittance of G-1, G-2 and G-3 solutions were also measured at different temperatures as shown in Figure 4. The transmittance of bioconjugates G-1, G-2 and G-3 started to decrease significantly when the temperature increased to 27 °C, 36 °C and 42 °C, respectively. At 35 °C, 43 °C and 47 °C, the light transmission of the bioconjugates solution decreased to near 20–30% and remained almost constant with increasing temperature due to the fact that the bioconjugates can self-assemble above the LCST to form vesicles or micelles and prevent its further precipitation and aggregation. Therefore, the LCSTs of bioconjugates G-1, G-2 and G-3 were 27 °C, 36 °C and 42 °C, respectively. As reported, the introduction of hydrophilic groups into the PEG polymer chain leads to an increase in the LCST of the polymer [37]. The LCST of hP(DEGMA-*co*-OEGMA)-*star*-PLZ is higher than that of the corresponding hyperbranched polymer hP(DEGMA-*co*-OEGMA) due to the coupling of hydrophilic lysozyme.

The bioconjugates hP(DEGMA-*co*-OEGMA)-*star*-PLZ is temperature-responsive and self-assembles at temperatures above its LCST. Since G-1, G-2 and G-3 are soluble in water at temperatures below their LCST, each polymer was dissolved in water at low temperatures and heated to 45 °C to observe their self-assembly behavior. Above 45 °C, the hyperbranched polymer nucleus becomes hydrophobic, while the lysozyme molecule is hydrophilic, which leads to the self-assembly of hyperbranched polymeric bioconjugates above LCST [38,39]. The self-assembly behavior and structure of these three bioconjugates were investigated by DLS and SEM. The hydrodynamic diameters (Dh) measured by DLS are shown in Figure 5, and their size and PDI are summarized in Table 5. The hydrodynamic diameter (Dh) was measured by DLS as shown in Figure 5, and its size and PDI are summarized in Table 5. Combined with Table 5, it can be seen that all three star-shaped biological adducts self-assemble to form narrower assemblies of PDI. SEM was used to further investigate their morphology, as shown in Figure 5b–d, G-1, G-2 and G-3 self-assemble into vesicles with concave and convex surfaces above their LCSTs. The dehydrated polymer chains collapsed due to hydrophobic interaction to form the inner vesicle core, while the hydrophilic lysozyme formed the outer vesicle shell. The sizes of G-1, G-2 and G-3 were in the ranges of 353–469 nm, 151–216 nm and 186–240 nm, respectively. It agrees with the DLS results.

To investigate the effect of the synthesis process on the biological activity of lysozyme, the biological activities of lysozyme, lysozyme-SH and the three protein-hybridized bioconjugates hP(DEGMA-*co*-OEGMA)-*star*-PLZ were tested at 25 °C. Lysozyme-sensitive Micrococcus lysis cells were used as reaction substrates for the lysozyme, lysozyme-SH and bioconjugates activity tests. The absorbance of the protein-hybridized bioconjugates hP(DEGMA-*co*-OEGMA)-*star*-PLZ at 450 nm versus reaction time under the test conditions at 25 °C is shown in Figure 6a. The bioactivity of lysozyme was defined as 100%, and the relative bioactivities of lysozyme on lysozyme-SH, bioconjugates G-1, G-2 and G-3 were determined as 99%, 93%, 87% and 82%, respectively (as shown in Figure 6b), indicating that the synthesis process had no significant effect on the lysozyme bioactivity.

To investigate the effect of synthesis and self-assembly processes on the conformation of lysozyme, the secondary structure of the lysozyme molecule on the bioconjugates hP(DEGMA-*co*-OEGMA)-*star*-PLZ hybridized by the three proteins was determined by CD spectroscopy, as shown in Figure 7. The structural curves of the lysozyme proteins were similar to those of lysozyme on the three bioconjugates, indicating that the modification of lysozyme with Traut’s reagent and the self-assembly process did not significantly affect the conformation of the lysozyme molecule.

## 4. Conclusions

Three thermosensitive hyperbranched polymers with different branching degrees and molecular weights were prepared by light-induced metal-free ATRP polymerization, and then a segment of DSMA monomer was further polymerized at the end of the hyperbranched polymer to introduce disulfide bonds to the hyperbranched polymer, and finally lysozyme modified with sulfhydryl groups was conjugated to the hyperbranched multiarm copolymers by the exchange reaction of sulfhydryl disulfide bonds to obtain protein-hybridized bioconjugates. The LCST of the bioconjugates was measured using turbidimetric method. With the grafting of lysozyme, the LCST of the bioconjugates shifted to a higher temperature compared to the hyperbranched polymer precursor, and the bioconjugates formed a nanoscale vesicular structure at temperatures higher than its LCST. During synthesis and self-assembly, the biological activity and secondary structure of the lysozyme on the bioconjugates remain largely unchanged compared to the natural lysozyme. The combination of hyperbranched polymers with biomolecules holds great promise for applications in biochemistry.

## Data Availability

Data will be made available on request.

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
