# Peer review of "Synthesis and Self-Assembly of Hyperbranched Multiarm Copolymer Lysozyme Conjugates Based on Light-Induced Metal-Free Atrp"

_nanomaterials, 2023, doi:10.3390/nano13061017_

Round 1

Reviewer 1 Report

In this study, the hyperbranched polymer hP(DEGMA-co-OEGMA) was first prepared by combining self-condensation vinyl polymerization (SCVP) with photo-induced metal-free atom transfer radical polymerization (ATRP), with 2-(2-bromo-2-methylpropanoyloxy) ethyl methacrylate (BMA) as inimer and Di(ethylene glycol) methyl ether methacrylate (DEGMA) and (oligoethylene glycol) methacrylate (OEGMA, Mn=300) as the copolymer monomer.

Some remarks:
1) describe the procedure for obtaining DLS results in the Experimental Section.
2) Figure 5a - why is the X axis on a logarithmic scale? It would be more logical to present graphs on a linear scale.
3) Is it possible to present similar results for SEM (meaning the distribution of particles from the diameter) - this would make the article more attractive to read.

Reviewer 2 Report

The paper is acceptable. There are a few changes that need to be made as listed below:

1)  page 2 line 3 and page 15 line 1 Capitalize the word "three" in the first word in the sentence. Also in Page 9 line 3 capitalize "lysozyme."

2) space in line 7 page 3.

3) Can you make illustrations of each of the monomers and how they assemble in the larger polymers. Can you make a table of illustrations of  individual monomers and how they form polymers?

4) Can you look at the computational works of Prof. Petr Kral? He has done many studies of branched and linear polymers. Also there s a publication by Atomically precise organomimetic cluster nanomolecules assembled via perfluoroaryl-thiol SNAr chemistry  by E. Qian et al looking into boron based clusters. Please cite this work as well as others done by Prof. Kral.
